# Real-World Data of Trastuzumab Deruxtecan for Advanced Gastric Cancer: A Multi-Institutional Retrospective Study

**DOI:** 10.3390/jcm11082247

**Published:** 2022-04-17

**Authors:** Toshihiko Matsumoto, Shogo Yamamura, Tatsuki Ikoma, Yusuke Kurioka, Keitaro Doi, Shogen Boku, Nobuhiro Shibata, Hiroki Nagai, Takanobu Shimada, Takao Tsuduki, Takehiko Tsumura, Masahiro Takatani, Hisateru Yasui, Hironaga Satake

**Affiliations:** 1Department of Medical Oncology, Kobe City Medical Center General Hospital, 2-1-1, Minatojimaminamimachi, Kobe 6500047, Japan; yamamura_shogo@kcho.jp (S.Y.); hiroki_nagai@kcho.jp (H.N.); hyasui@kcho.jp (H.Y.); takeh1977@gmail.com (H.S.); 2Cancer Treatment Center, Kansai Medical University, 2-3-1, Shinmachi, Hirakata 5731191, Japan; shogen0820@gmail.com (S.B.); shibanob.kmu@gmail.com (N.S.); 3Department of Clinical Oncology, Osaka Red Cross Hospital, 5-30, Tenoji-ku, Fudegasakicho, Osaka 5438555, Japan; quouqbnonb@gmail.com (T.I.); kdoi0160@osaka-med.jrc.or.jp (K.D.); shimadatakanobu@osaka-med.jrc.or.jp (T.S.); t-tsumura@osaka-med.jrc.or.jp (T.T.); 4Department of Internal Medicine, Himeji Red Cross Hospital, 1-12-1, Shimoteno, Himeji 6708540, Japan; sagc3231572@gmail.com (Y.K.); tsuzuo@gmail.com (T.T.); takatanimsh@gmail.com (M.T.); 5Department of Medical Oncology, Kochi Medical School, Nankoku 7838505, Japan

**Keywords:** gastric cancer, trastuzumab deruxtecan, T-DXd, chemotherapy, real-world data

## Abstract

**Simple Summary:**

Gastric cancer has the fifth highest incidence among cancers in the world, and it causes the third most cancer-related deaths. In this study, data from patients with *HER2-*positive advanced gastric cancer who received trastuzumab deruxtecan (T-DXd) were analyzed. Pre-administration of immune checkpoint inhibitors and a sufficiently long trastuzumab-free interval may be predictive factors of T-DXd efficacy.

**Abstract:**

Trastuzumab deruxtecan (T-DXd) has shown promising efficacy against *HER2-*positive advanced gastric cancer (AGC). However, data on its real-world efficacy in AGC patients are insufficient, and the predictive marker of T-DXd is unclear. In this multi-center retrospective study, we collected clinical information of 18 patients with *HER2*-positive AGC who received T-DXd after intolerant or refractory responses to at least two prior regimens and analyzed predictive factors. The median age was 71 years (range: 51–85), 13 men were included, and ECOG performance status (PS): 0/1/2/3 was 9/6/2/1. A total of 11 patients (61%) received prior immune checkpoint inhibitors (ICIs), 14 patients were *HER2* 3+, and 4 patients were *HER2* 2+/FISH positive. The median trastuzumab (Tmab)-free interval was 7.7 months (range: 2.8–28.6). The overall response rate was 41%, and the disease control rate was 76%. Median progression-free survival (PFS) was 3.9 months (95% CI: 2.6–6.5), and median overall survival (OS) was 6.1 months (95% CI: 3.7–9.4). PFS (6.5 vs. 2.9 months, *p* = 0.0292) and OS (9.2 vs. 3.7 months, *p* = 0.0819) were longer in patients who received prior ICIs than in those who had not. PFS (6.5 vs. 3.4 months, *p* = 0.0249) and OS (9.4 vs. 5.7 months, *p* = 0.0426) were longer in patients with an 8 month or longer Tmab-free interval. In patients with ascites, PFS (6.5 vs. 2.75 months, *p* = 0.0139) and OS (9.4 vs. 3.9 months, *p* = 0.0460) were shorter. T-DXd showed promising efficacy in *HER2*-positive AGC patients in a real-world setting. Pre-administration of ICIs and a sufficient Tmab-free interval may be predictive factors of T-DXd efficacy.

## 1. Introduction

Gastric cancer is known to be one of the most common cancers, and the frequency of occurrence is the fifth highest among cancers, and cancer-related deaths are the third highest [1]. Globally, multimodal treatments including surgery and perioperative/adjuvant chemotherapy have improved the treatment outcome of patients with early-stage disease. However, the treatment outcomes of patients with advanced gastric cancer (AGC) are still poor.

Recently, targeted treatments have demonstrated a significant benefit in gastric cancer. Human epidermal growth factor receptor-2 (*HER2*) is currently the most useful therapeutic biomarker for patients with AGC, because *HER2* overexpression in AGC cases ranges from 6% to 30% [2]. The phase III trial of trastuzumab (ToGA trial) for *HER2-*positive AGC patients demonstrated a significant efficacy as a first-line chemotherapy [3]. In the ToGA trial, chemotherapy plus trastuzumab (Tmab) significantly prolonged median overall survival (OS) (13.8 vs. 11.1 months, hazard ratio (HR):0.74, *p* = 0.0046) compared with chemotherapy alone. In an exploratory post hoc analysis, OS improved more (16.0 vs. 11.8 months, HR:0.65, *p* = 0.036) in the *HER2*-positive group (defined as immunohistochemical(IHC) 2+ and a FISH-positive result, or IHC 3+, regardless of FISH status). Based on these results, chemotherapy plus Tmab became the standard chemotherapy for chemo-naïve *HER2*-positive AGC. 

Trastuzumab deruxtecan (T-DXd) is an antibody–drug conjugate composed of an anti-*HER2* antibody, a cleavable tetrapeptide-based linker, and a cytotoxic topoisomerase inhibitor [4]. In a dose expansion phase I study, T-DXd showed 43.2% of confirmed RR, 5.6 months of PFS, and 12.8 months of OS in *HER2*-positive AGC [5]. Following this trial, a randomized phase II trial (DESTINY-Gastric01) was conducted. This study was open label study compared with the physician’s choice therapy (docetaxel or irinotecan), and the primary endpoint was the response rate (RR). T-DXd showed significantly higher RR (51% vs. 14%, *p* < 0.001) and prolonged OS (12.5 vs. 8.4 months, hazard ratio (HR): 0.59, *p* = 0.01) and median progression-free survival (PFS) (5.6 vs. 3.5 months, HR:0.47) [6]. Based on these results, T-DXd was approved in Japan, the United States, and the European Union.

There were no *HER2*-positive AGC patients with poor general health conditions in the clinical trial [3,5]. In the DESTINY-Gastric01 trial, 50% of patients had PS 0 and 50% had PS 1, and the proportion of cases with ascites was unknown. In the real world, T-DXd is also administered to AGC patients with poor PS, the elderly, and ascites. There are no efficacy and safety data for T-DXd in patients with those conditions. In the DESTINY-GC01 study, the pre-specified subgroup analysis reported that *HER2* immunohistochemical (IHC) analysis score 3+ cases showed a better response rate than *HER2* IHC 2+ cases (58% vs. 29%), but other predictive factors and prognosis factors are not yet clear. Therefore, we planned this study to explore the efficacy, toxicity, and prognostic factors of T-DXd in the real world.

## 2. Materials and Methods

### 2.1. Patients

This was a multi-institution study. The subjects were AGC patients received T-DXd between August 2018 and January 2021 at four institutions. Clinical information was retrospectively collected based on the electronic medical records. This study was performed in accordance with institutional and national standards on human experimentation, as confirmed by the ethics committee of all participating institutions and with the principles of the Declaration of Helsinki.

The inclusion criteria of this study were as follows: (1) unresectable gastric cancer, (2) histologically proven gastroesophageal adenocarcinoma, (3) refractory or intolerant to at least two regimens, and (4) *HER2* levels were documented as high (score of 3+ on immunohistochemical (IHC) analysis or score of 2+ and positive results on fluorescence in situ hybridization (FISH)). This study was approved by the Institutional Review Board of the Kobe City Medical Center General Hospital (Examination number: zn211018) and all other institutions. All participants were given with opportunities to decline agreement for this study by the ‘opt-out’ option on our hospital homepage.

### 2.2. Treatment

The patients were administrated T-DXd 6.4 mg/kg infusion every 3 weeks until disease progression or intolerance.

### 2.3. Evaluation and Statistical Analysis

Tumor response was evaluated by the Response Evaluation Criteria in Solid Tumors (RECIST) version 1.1. Adverse events were defined with the Common Terminology Criteria for Adverse Events (CTCAE) version 4.1. Analysis of OS and PFS were performed using the Kaplan–Meier method. OS was defined as the time from date of initiation of treatment with T-DXd until death. Patients who were alive or for whom data were missing at the data cut-off point were censored. PFS was defined as the time from the date of initiation of treatment with T-DXd until disease progression or death from any cause. Censored cases were defined as the patients with no information of tumor progression. Univariate Cox proportional hazards models were performed to investigate the risk factors correlated with PFS and OS. Adverse events were evaluated by the Common Terminology Criteria for Adverse Events ver. 4.0. JMP version 12 (SAS Institute Inc., Cary, NC, USA), which was used for statistical analysis.

## 3. Results

Between August 2018 and January 2021, 18 patients were treated T-DXd after the failure of at least two regimens. The patients’ backgrounds are presented in Table 1. All patients had a metastatic site. The median age was 71 (range: 51–85) and 72% were male. There were 15 patients with PS 0 or 1 and 3 patients with PS 2 or 3. In total, 10 patients (56%) had a diffuse-type histology. A total of 14 patients (78%) had *HER2* IHC 3+, whereas 4 patients (22%) were *HER2* IHC 2+/FISH positive. Overall, 15 patients (83%) had two or more organ metastasis, 11 patients (61%) had peritoneal dissemination, and 7 (39%) had liver metastasis. Furthermore, 14 patients (78%) experienced partial response (PR) in a prior trastuzumab (Tmab)-containing regimen; the median Tmab-free interval was 7.7 months (range: 2.8–28.6 months).

### 3.1. Efficacy

The effect evaluation by the image was possible with 17 patients. PR was observed in 41%, and 35% of patients achieved stable disease (SD); the response rate (RR) was 41% and the disease control rate (DCR) was 76%. The median follow-up time was 5.8 months (range: 0.5–30.5 months) among censored cases. The median PFS was 3.9 months (95% confidence interval (CI), 2.6–7.7), the median OS was 6.5 months (95% CI, 3.7–10.6), and the 1-year survival rate was 16.7% (Figure 1).

Eleven patients (61%) had received prior immune checkpoint inhibitors (ICIs). The median PFS (6.5 vs. 2.9 months, *p* = 0.0292) was longer, whereas the median OS (9.2 vs. 3.7 months, *p* = 0.0819) tended to be better in patients who were treated with prior ICIs than in those who were not. In addition, RR (60% vs. 14%, *p* = 0.0595) and DCR (90% vs. 57%, *p* = 0.1147) tended to be better in patients who received prior ICIs than in those who did not (Figure 2a,b and Figure 3).

The median Tmab-free interval (TFI) was 7.7 months, and nine patients (50%) had a TFI of 8 months or longer. PFS (6.5 vs. 3.4 months, *p* = 0.0249) and OS (9.4 vs. 5.7 months, *p* = 0.0426) were significantly better in patients with a TFI of 8 months or longer than in patients with a TFI of less than 8 months. In addition, RR (62% vs. 22%, *p* = 0.0878) tended to be better and DCR (100% vs. 56%, *p* = 0.0129) was better in patients with a TFI of 8 months or longer than in patients with a TFI of less than 8 months. (Figure 2c,d and Figure 3). No significant relationship was observed between prior ICI administration and TFI (*p* = 0.2151).

Median Tmab PFS was 7.1 months (range: 0.93–22.3). PFS of T-DXd was significantly better in patients with Tmab PFS > 7 months than in those with Tmab PFS < 7 months (5.9 vs. 3.4 months, *p* = 0.0426). RR (50% vs. 14%, *p* = 0.116) and DCR (80% vs. 57%, *p* = 0.3105) also showed a good tendency in patients with Tmab PFS > 7 months than in those with Tmab PFS < 7 months (Appendix A).

Eleven patients (61%) had peritoneal dissemination, whereas eight patients (44%) had ascites. PFS (6.5 vs. 2.75 months, *p* = 0.0139) and OS (9.4 vs. 3.9 months, *p* = 0.0460) were shorter in patients with ascites than in patients without ascites (Figure 4). Further, a decrease in ascites was observed in 25% of the patients with ascites after T-DXd administration. 

Seventeen patients (94%) discontinued T-DXd, and six patients (35%) received post-treatment. Four patients received nivolumab, and two patients received paclitaxel + ramucirumab.

### 3.2. Safety

Table 2 shows adverse events of this study. Most of the grade 3 or higher adverse events were neutropenia (17%), anemia (11%), anorexia (6%), and nausea (6%). Grade 2 pneumonia and grade 2 heart failure were observed in one patient each (6% each). Treatment was discontinued in 17 patients: 16 patients discontinued treatment due to tumor progression, whereas 1 patient discontinued treatment due to grade 2 heart failure. No treatment-related deaths occurred. 

## 4. Discussion

In the DESTINY-Gastric01 study, T-DXd showed promising efficacy as the third- or later-line treatment for *HER2-*positive AGC patients [6]. In our real-world study, T-DXd showed a 41% RR, and the median PFS and OS were 3.9 months and 6.5 months, respectively. Similar PFS and RR were observed despite the inclusion of 17% PS2 in three cases in the present study. This result suggests that T-DXd is a useful treatment for *HER2*-positive AGC patients not only in clinical trials, but also in real-world patients.

This study showed that PFS was longer in patients who received prior ICIs than that in other patients, whereas OS tended to be longer. This result has not been reported in previous studies, including the DESTINY-GC01 trial. These results suggest the possibility of a synergistic action between T-DXd and ICIs. Osa et al. previously reported that binding to nivolumab and T cells was observed more than 20 weeks after the last dose, regardless of the total dose of nivolumab [7]. Previous preclinical studies showed that tumor recognition by T cells is enhanced by T-DXd. T-DXd increases tumor infiltrative CD8 + T cells, and the expression of PD-L1 and major histocompatibility complex class I in tumor cells is also enhanced by T-DXd [8]. Therefore, several studies are currently underway to explore the efficacy of a combination of multiple T-DXd and immune checkpoint inhibitors. In our study, PFS was longer in patients with ICIs free interval <20 weeks than in patients with ICIs free interval >20 weeks (Appendix A). This result suggests that combination or sequential therapy with T-DXd and ICIs may be useful for AGC patients. 

Further, in our study, the efficacy of T-DXd was significantly better in patients with a TFI of 8 months or longer. There are no data on the correlation between TFI and the effect of T-DXd. Makiyama et al. reported a randomized, phase II study exploring efficacy of trastuzumab beyond progression for AGC patients who were *HER2* positive (WJOG7112G: T-ACT Study) [9]. In the T-ACT study, no additional effect of Tmab on paclitaxel was observed in the entire population. However, in patients with a long TFI (30 or more days), the administration of Tmab showed significant improvement of PFS (HR, 0.45; 95%CI, 0.21–0.96; interaction test, *p* = 0.022). Makiyama et al. also reported that the *HER2* copy number is high in cases with long Tmab-free intervals. Pietrantonio et al. reported that *HER2* overexpression was decreased in 32% of patients who previously received trastuzumab-based chemotherapy [10], and Wang et al. reported that patients with acquired resistance of Tmab have reduced *HER2* somatic copy number alterations [11]. Similarly, the therapeutic effect of T-DXd may also be enhanced by prolonging the TFI and increasing the *HER2* copy number. In the DESTINY-GC01 trial, patients with high plasma *HER2* copies number and high *HER2* extracellular domain (ECD) tended to be more effective with T-DXd [12]. If TFI is extended, *HER2* amplification may occur, and the effect of T-DXd may be enhanced. However, data on the relationship between TFI and T-DXd efficacy are limited, and further investigations are needed. In addition, in our study, the efficacy of T-DXd tended to be higher in patients with longer PFS after prior therapies with a Tmab combination. Since T-DXd is used after third-line therapy, re-biopsy is often difficult in clinical practice. Therefore, in cases where re-biopsy is not possible, PFS and TFI of pretreatment Tmab may be useful to predict the efficacy of T-DXd. 

Moreover, exploratory cohorts in DESTINIY-Gastric01 trial, T-DXd improved RR (26.3% va 9.5%) and PFS (4.4 vs. 2.8 months) in *HER2* IHC 1+ or *HER2* IHC 2+ and FISH-negative patients [12]. This result suggested that T-DXd had modest efficacy in *HER2* low AGC. Although this result was only a small number exploratory analysis, and further prospective studies for *HER2*-low AGC patients are warranted.

In a real-world setting, many cases of AGC have peritoneal dissemination or ascites. The DESTINY-Gastric 01 study does not reveal the proportion of cases with peritoneal dissemination or ascites, nor does it reveal efficacy data. In our study, 61% had peritoneal dissemination and 44% had ascites. In patients with ascites, PFS and OS were significantly worse than patients without ascites. However, in patients with ascites, only 25% of patients showed a decrease in ascites. This result is similar to the therapeutic effect of nivolumab [13]. In our study, there was no difference in the frequency of serious toxicity by T-DXd with or without ascites. These results indicate that, although T-DXd shows a modest effect for *HER2*-positive gastric cancer with ascites, careful use is required.

The mechanism of resistance after T-DXd administration is not yet clear. Reports of treatment with trastuzumab suggest that *ERBB2* exon 16 and mutations in the receptor tyrosine kinase, *PI3K,* and *RAS* pathways are related to resistance of anti-*HER2* therapy [14]. In our study, genome profiling using next-generation sequencing was conducted in only two cases before and after T-DXd administration. In one case, the *HER2* amplification observed before treatment disappeared after progressive disease and the *PIK3CA E542K* mutation appeared. This suggests that a known Tmab resistance mechanism may be involved in the resistance to T-DXd, but there are still few reports of biomarkers involved in T-DXd resistance, including changes in *HER2* amplification. Further research is needed.

## 5. Conclusions

Some limitations were found in our trial. First, the study was a retrospective study with small sample size of only Japanese patients, but it nevertheless indicated that T-DXd has promising effects and is tolerable as a third- or later-line therapy for *HER2*-positive AGC patients. It is suggested that the administration history of immune checkpoint inhibitors, short ICI intervals, long TFI, and long-term tumor control with Tmab may be predictors of the therapeutic effect of T-DXd. However, in patients with ascites, our study suggests that the effect may not be sufficient. Further investigations to explore efficacy, safety, and more convenient predictive factors of T-DXd are needed.

## Figures and Tables

**Figure 1 jcm-11-02247-f001:**
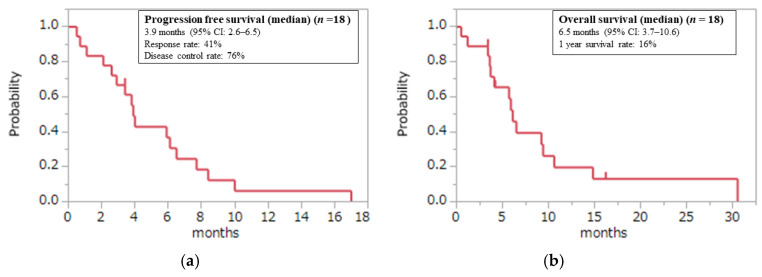
(**a**) Progression-free survival (PFS) and (**b**) overall survival (OS) among all study population.

**Figure 2 jcm-11-02247-f002:**
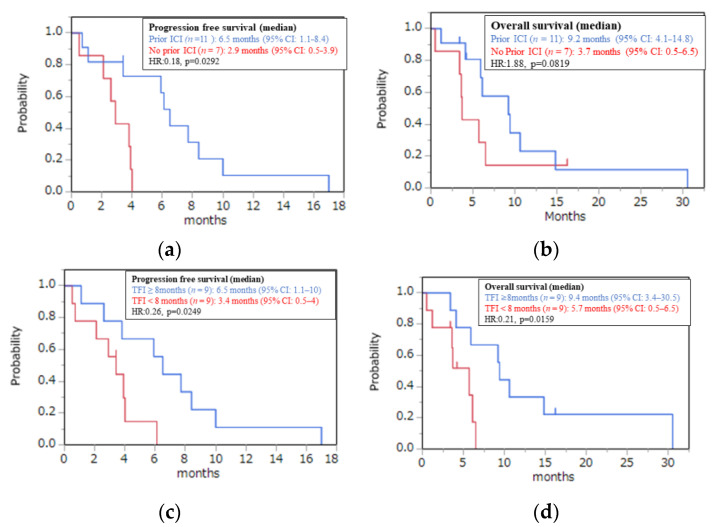
(**a**) PFS and (**b**) OS among study participants classified by prior therapy. Red line: No prior immune checkpoint inhibitor (ICI); blue line: prior ICI; Kaplan–Meier plots of (**c**) PFS and (**d**) OS among study participants. Red line: Tmab-free interval (TFI) < 8 months; blue line: TFI ≥ 8 months.

**Figure 3 jcm-11-02247-f003:**
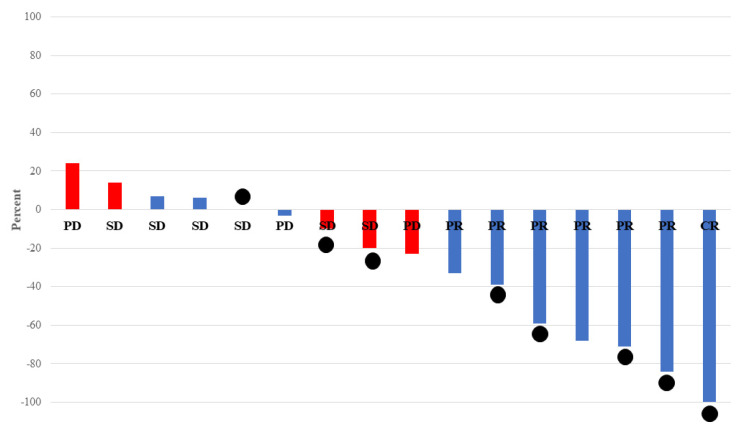
Best rate of change from baseline in total measurable tumor diameter. Blue: Prior use of ICI (*n* = 10): (RR: 60%), Red: no prior use of ICI (*n* = 6): (RR: 14%), ●: Tmab-free interval over 8 months (*n* = 8): (RR: 75%).

**Figure 4 jcm-11-02247-f004:**
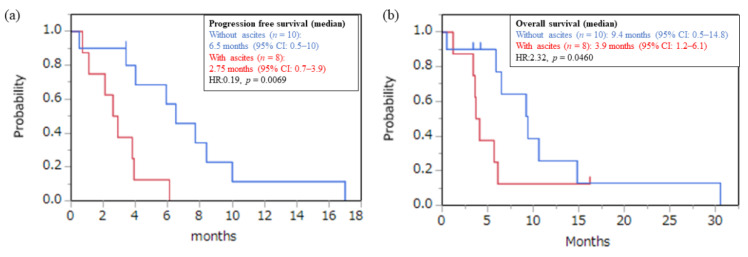
(**a**) PFS and (**b**) OS curve classified by ascites. Red line: with ascites; blue line: without ascites.

**Table 1 jcm-11-02247-t001:** Patients’ characteristics of this study.

		All (*n* = 18)
Age	Median (range)	71 (51–85)
Sex	Male	13 (72%)
ECOG PS	0	9 (50%)
	1	6 (33%)
	2/3	3 (17%)
Primary site	Gastric	14 (78%)
	EGJ	4 (4%)
Histology	Diffuse type	10 (56%)
	Intestinal type	8 (44%)
*HER2* status	IHC 3+	14 (78%)
	IHC 2+, FISH (+)	4 (22%)
Prior gastrectomy	Yes	9 (50%)
Number of metastatic sites	≥2	15 (83%)
Liver metastasis	Yes	7 (39%)
Peritoneal dissemination	Yes	11 (61%)
Ascites	Yes	8 (44%)
Massive ascites	Yes	7 (39%)
Measurable lesion	Yes	17 (94%)
Number of prior regimens	2	6 (33%)
	Over 2	12 (67%)
Prior 5-FU	Yes	18 (100%)
Prior Platinum	Yes	18 (100%)
Prior taxane	Yes	16 (89%)
Prior irinotecan	Yes	3 (20%)
Prior ramucirumab	Yes	16 (89%)
Prior immune check point inhibitors	Yes	11 (61%)
Efficacy of Prior Tmab regimen	PR/SD/PD	14/3/1
Response rate of prior Tmab	PR	14 (78%)
Tmab Free interval (months)	Median (range)	7.7 (2.8–28.6)

ECOG PS, Eastern Cooperative Oncology Group performance status; EGJ, gastroesophageal junction carcinoma; *HER2*, human epidermal growth factor receptor; 5-FU, 5-fluorouracil; Tmab, trastuzumab; PR, partial response; SD, stable disease; PD, progressive disease.

**Table 2 jcm-11-02247-t002:** Distribution of drug-related adverse events.

	ALL	G1/2	G3/4
Neutropenia	5 (28%)	2 (11%)	3 (17%)
Anemia	6 (33%)	4 (22%)	2 (11%)
Platelet decreased	2 (11%)	2 (11%)	0
Fatigue	11 (61%)	11 (61%)	0
Anorexia	7 (39%)	6 (33%)	1 (6%)
Nausea	5 (28%)	4 (22%)	1 (6%)
Diarrhea	3 (20%)	3 (20%)	0
Heart failure	2 (11%)	2 (11%)	0
Pneumoniae	1 (6%)	1 (6%)	0
Rush	1 (6%)	1 (6%)	0
Constipation	1 (6%)	1 (6%)	0
Fever	1 (6%)	1 (6%)	0

## Data Availability

All the data and materials supporting the conclusions are included in the main paper. The datasets used in the current study are available from the corresponding author upon request.

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
