# Peer review of "Real-World Data of Trastuzumab Deruxtecan for Advanced Gastric Cancer: A Multi-Institutional Retrospective Study"

_jcm, 2022, doi:10.3390/jcm11082247_

Round 1
Reviewer 1 Report
The study by Toshihiko Matsumoto et al. “Real world data of trastuzumab deruxtecan for advanced gastric cancer: A multi-institutional retrospective study ” is a retrospective study on HER2-positive AGC patients.
The text is quite simple and deals with an important issue: T- DXd treatment strategy for HER2-positive in third- or later-line therapy.
As mentioned by the authors , the study had some limitations as well as, the small sample size.
I would suggest to deepen , if possible, the analysis of tumor-infiltrating CD8+ T cells.
Minor points:
- the authors should improve the quality of the tables
- the authors should better describe clinical trials in the introduction.
Author Response
Thank you very much for reviewing our manuscript and offering valuable advice.
We have addressed your comments with point-by-point responses, and revised the manuscript accordingly.
Minor points:
- the authors should improve the quality of the tables
→Thank you for your suggestion. We have adjusted the table to be as easy to read as possible. Please check it.
- the authors should better describe clinical trials in the introduction.
→Thank you for your suggestion. We added more information of clinical trial information. (Line58-69,red lesion)
Reviewer 2 Report
Authors provided interesting insights into HER2+ gastric cancer patients treated with T-DXd and immune checking blockade (ICB), which could be beneficial for clinical management.
Author Response
Thank you very much for reviewing our manuscript and offering valuable advice.
Reviewer 3 Report
Matsumoto et al analyzed the efficacy of Trastuzumab deruxtecan (T-DXd) as treatment of HER2-positive advanced gastric cancer that had progressed after at least two previous therapeutic regimens including trastuzumab. This multi-institutional retrospective study has obvious limitations but a) confirms the promising efficacy of T-DXd in a real-world setting b) highlights that administration of immune checkpoint inhibitors before T-DXd and a prolonged trastuzumab-free interval are associated with better T-DXd efficiency. On the contrary, the presence of ascites may be predictive of worse response to T-DXd treatment.
Comments-questions:
- Please define Advanced Gastric Cancer (AGC) term. I assume it includes locally advanced and metastatic gastric cancer, is it right?
- Does the cohort include only gastric or both gastric and gastroesophageal junction adenocarcinoma patients?
- Please expand the introduction section with more information about the use of trastuzumab as first-line choice in HER2+ AGC and the advantages of T-DXd as a third-line treatment in AGC, a cancer with heterogeneous HER2 expression.
- Please clarify in the abstract that T-DXd was administered after 2+ therapeutic regimens and not as first line choice.
- Please define ECOG.
- Line 58 Which clinical trial?
- Table 1: Please define PR, SD, PD
8: Table 1: Please provide information about the dimension and localization (if both gastric and gastroesophageal are included in the study) of measurable tumors.
- Figure 3 (line 150): is n=7 right for red (non ICI)? Only 6 red bars are depicted in the figure
- Figures 2,4 and suppl figures: Please provide Hazard Ratios
- Suppl figure 1: Please correct numbers because total n=19 and not 18
- Suppl figure 2 is not mentioned in the manuscript.
- Is the limited ethnic diversity another limitation of the study?
- DESTINY-Gastric01 reported that HER2 3+ patients had better response to T-DXd. Do you see a similar trend if you perform a HER2 3+ vs 2+ subgroup analysis?
- HER2 expression has been reported to be down-regulated after initial therapy with Tmab in several studies*. What happens with T-DXd? Does it also work on lower levels of HER2 expression (1+) or only in cases where long Tmab free intervals presumably keep HER2 expression high? Please expand on this in the discussion section.
* Wang D-S, Liu Z-X, Lu Y-X, et al. Liquid biopsies to track trastuzumab resistance in metastatic HER2-positive gastric cancer. Gut 2019;68:1152-1161.
Pietrantonio F, Caporale M, Morano F, et al. HER2 loss in HER2-positive gastric or gastroesophageal cancer after trastuzumab therapy: implication for further clinical research. Int J Cancer 2016;139:2859-2864.
Author Response
Thank you very much for reviewing our manuscript and offering valuable advice.
We have addressed your comments with point-by-point responses, and revised the manuscript accordingly.
|
 Minor comments |
  |
|
|
No. 1 |
Please define Advanced Gastric Cancer (AGC) term. I assume it includes locally advanced and metastatic gastric cancer, is it right? |
Thank you for important indication. AGC includes both of locally advanced and metastatic gastric cancer. But, this study includes only metastatic gastric cancer. We add the sentence( line 118, red section) |
|
No. 2 |
Does the cohort include only gastric or both gastric and gastroesophageal junction adenocarcinoma patients? |
Thank you for important indication. This study included both gastric and EGJ. We added the site of the primary lesion to the table. 1. |
|
No.3
|
Please expand the introduction section with more information about the use of trastuzumab as first-line choice in HER2+ AGC and the advantages of T-DXd as a third-line treatment in AGC, a cancer with heterogeneous HER2 expression. |
Thank you for your important suggestion. We added the information to line51-68 (red section) |
|
No.4 |
Please clarify in the abstract that T-DXd was administered after 2+ therapeutic regimens and not as first line choice. |
Thank you for important indication. We added this information to line 117-118(red section) |
|
No.5 |
Please define ECOG.
|
Thank you for important indication. We added this information to table.1. |
|
No.6 |
Line 58 Which clinical trial? |
Thank you for important indication. We added this information to line 69. |
|
No.7 |
Table 1: Please define PR, SD, PD |
Thank you for important indication. We added this information to table.1. |
|
No.8 |
Table 1: Please provide information about the dimension and localization (if both gastric and gastroesophageal are included in the study) of measurable tumors. |
Thank you for important indication. We added this information to table.1. |
|
No.9 |
Figure 3 (line 150): is n=7 right for red (non ICI)? Only 6 red bars are depicted in the figure |
Thank you for important indication. We have corrected the wrong number in the Figure.3. |
|
No.10 |
Figures 2,4 and suppl figures: Please provide Hazard Ratios |
Thank you for important indication. We added date of Hazard ratio to Figure.2 and 4. |
|
No.11 |
Suppl figure 1: Please correct numbers because total n=19 and not 18
|
Thank you for important indication. We have corrected the wrong number in the Suppl figure.1. |
|
No.12 |
Suppl figure 2 is not mentioned in the manuscript. |
Thank you for important indication. We added the mention of Suppl fig.2 to line 224. |
|
No.13 |
Is the limited ethnic diversity another limitation of the study? |
Thank you for important indication. Both the DESTINY Gastric 01 study and our study are for Asians only. We added it as a limitation.(line 276, red section) |
|
No.14 |
DESTINY-Gastric01 reported that HER2 3+ patients had better response to T-DXd. Do you see a similar trend if you perform a HER2 3+ vs 2+ subgroup analysis? |
Thank you for important indication. In our study, there were only 4 cases of IHC2 +. We analyzed the effects and found no significant difference between IHC2 + and 3+ cases. The number of cases is too small, 4 cases, so it is not described in this paper. |
|
No.15 |
HER2 expression has been reported to be down-regulated after initial therapy with Tmab in several studies*. What happens with T-DXd? Does it also work on lower levels of HER2 expression (1+) or only in cases where long Tmab free intervals presumably keep HER2 expression high? Please expand on this in the discussion section. * Wang D-S, Liu Z-X, Lu Y-X, et al. Liquid biopsies to track trastuzumab resistance in metastatic HER2-positive gastric cancer. Gut 2019;68:1152-1161.
Pietrantonio F, Caporale M, Morano F, et al. HER2 loss in HER2-positive gastric or gastroesophageal cancer after trastuzumab therapy: implication for further clinical research. Int J Cancer 2016;139:2859-2864. |
Thank you for important indication. We totally agree with you and believe that the resistance mechanism after T-DXd and the effectiveness of T-DXd in HER2-low positive cases are very important. The data is still limited at this point, but We added it to the discussion.(line 249-253, line 271-272,red section.) and cited these 2 papers. |
